

# Thermokarst lakes disturb the permafrost structure and stimulate through-talik formation in the Qinghai–Tibet Plateau, China: A hydrogeophysical investigation

Xianmin Ke[1], Wei Wang[2,3,4], Fujun Niu[5], Zeyong Gao[6], Wenkang Huang[7], Huake Cao[8]

[1]College of Energy and Power Engineering, Lanzhou University of Technology, Lanzhou 730050, China
[2]School of Water and Environment, Chang'an University, Xi'an, 710054, Shaanxi, China
[3]Key Laboratory of Eco-hydrology and Water Security in Arid and Semi-arid Regions of the Ministry of Water Resources, Chang'an University, Xi'an 710054, China
[4]Key Laboratory of Subsurface Hydrology and Ecological Effect in Arid Region of the Ministry of Education, Chang'an University, Xi'an, 710054, Shaanxi, China
[5]School of Environment and Geographic Sciences, Shanghai Normal University, Shanghai, 200234, China
[6]State Key Laboratory of Frozen Soil Engineering, Northwest Institute of Eco-Environmental and Resources, CAS, Lanzhou, 730000, China
[7]Guangdong Research Institute of Water Resources and Hydropower, Guangzhou, 510630, Guangdong, China
[8]College of Geological Engineering and Geomatics, Chang'an University, Xi'an, 710054, Shaanxi, China

*Correspondence to*: Xianmin Ke (kexianmin@lut.edu.cn), Wei Wang (wangweichd@chd.edu.cn)

**Abstract.** Thermokarst lakes are widely distributed in the Qinghai–Tibet Plateau (QTP) and continuously disturb the permafrost structure. Investigating the permafrost and sublake talik structures is crucial to assessing and predicting the fate of the ecosystem and engineering under climate warming. Until recently, measurements of the permafrost distribution are often limited to seasonally frozen soil or permafrost at a few borehole locations, and the detection of deep permafrost and sublake taliks in the QTP has rarely been attempted on larger scales. Here, a synergistic application of electrical resistivity tomography, transient electromagnetic method, and borehole temperature measurement was used to investigate the permafrost and sublake talik structures in a thermokarst lake region of the QTP. The results showed that the maximum lower limit depths of the permafrost and active layer were determined to be 84–100 m and 0.9–4.0 m, respectively. Sub- and supra-permafrost water continuously erode the base and top plate of the permafrost, thereby reducing its thickness and disturbing its structure. Moreover, thermokarst lakes (unofficially named lakes BLH–A, B, and C) thaw the surrounding permafrost and form three through-taliks below them. These findings can help understand the interaction between thermokarst lakes and permafrost and optimize cryohydrogeologic models that can predict the evolution of permafrost and thermokarst lakes in similar cold regions.

## 1 Introduction

Permafrost is a special type of sediment that remains in the frozen state for several years (Gao and Coon, 2022) and covers approximately a quarter of the land surface in the Northern Hemisphere (Liu et al., 2021; Obu, 2021). The Qinghai–Tibet Plateau (QTP), which has the largest body of permafrost at high elevations and low latitudes (Zhou et al., 2022a; Sun et al.,



2023), is crucial for maintaining the global climate and ecological environment (Yao et al., 2012; Liu et al., 2023a). However, climate warming caused rapid thawing of permafrost in this region, which is expected to continue throughout the 21st century

(Zhang et al., 2022; Zhang et al., 2023a). Thermokarst lakes caused by permafrost degradation are a common natural landscape, widely distributed in permafrost regions (Li et al., 2021). The surface area and number of thermokarst lakes caused by permafrost degradation in the QTP have been increasing (Luo et al., 2022). Thermokarst lakes are believed to serve as heat sources that thaw the surrounding permafrost (Peng et al., 2021). Where thermokarst lakes are present, through-taliks may exist and provide flow pathways that accelerate permafrost degradation (Ke et al., 2023a), lead to the loss of supra-permafrost

water resources (Yoshikawa and Hinzman, 2003), change the hydrological cycle (Li et al., 2021), affect groundwater quality (Bouchard et al., 2011), and reconfigure the geomorphological landscape (Veremeeva et al., 2021). Additionally, thermokarst lakes have a strong thermal effect on adjacent infrastructure, thus affecting their stability and inducing serious engineering diseases (Guo et al., 2016; Wen et al., 2018; Liu et al., 2023b). As such, investigating the distribution of permafrost and sublake taliks is valuable for understanding and predicting the fate of the ecosystem, hydrological environment, and engineering

projects in the QTP under climate warming.

The thermal state and thickness of permafrost are typically determined from core drilling (Noetzli et al., 2021), ground temperature measurement (GTM) (Uhlemann et al., 2023), and modeling (Li et al., 2014). Wu et al. (2010) showed that the permafrost thickness in the QTP ranged from 10 to 312 m, whereas the maximum thicknesses obtained from borehole measurements and geothermal gradient estimation were 128 m (Tanggula Mountains) and 312 m (Fenghuo Mountains),

respectively. The permafrost thicknesses in alpine, hilly, and plain or river valleys are >200, 60–130, and <60 m (Cheng et al., 2019), respectively. Lin et al. (2016) revealed the thermal state and extinction process of the permafrost within 50 m below a thermokarst lake using nearly 10 years of ground temperature monitoring data. Core drilling and GTM for detecting permafrost and taliks are valuable but labor-intensive and point-scale (You et al., 2017), making it difficult to estimate the spatial distribution (larger than the point scale) of permafrost. Additionally, although it is easy to monitor shallow permafrost, it is

difficult to accurately detect permafrost structures with a large thickness because of the limitation of the drilling depth (Uhlemann et al., 2023). Numerical models play an important role in revealing the spatiotemporal evolution of permafrost and thermokarst lakes at large scales (Wellman et al., 2013; Li et al., 2014; Li et al., 2021). However, these models are based on some simplified assumptions (e.g., lake geometries are regular and constant, precipitation and evaporation are ignored, saturation models, etc.) and formation structure (permafrost and sublake taliks) that cannot be supported by actual

measurements, which imply a trade-off between model complexity and simulation accuracy (Lamontagne-Halle et al., 2020). To detect larger-scale (larger than borehole locations) and more accurate permafrost and sublake taliks structures for cryohydrogeologic modeling, convenient and dense spatial measurement technologies are required.

Geophysical prospecting methods (GPMs) are noninvasive techniques for obtaining subsurface physical information, and they can help overcome the above limitations and improve our understanding of multiscale subsurface structures (Binley et al.,

2015). Electrical resistivity tomography (ERT) (Briggs et al., 2017; Hornum et al., 2021), transient electromagnetic (TEM) (Creighton et al., 2018), ground penetrating radar (GPR) (Terry et al., 2020), and nuclear magnetic resonance (NMR) (Keating



et al., 2018) can be used for detecting permafrost and taliks; these techniques are respectively based on the differences in the electrical resistivity (ER), dielectric permittivity, and electromagnetic excitation of the protons in hydrogen atoms between frozen and thawed sediments. GPMs are often used in combination to determine the permafrost distribution and improve the accuracy of the results. For example, the distribution of permafrost and taliks has been determined by combining GPR and ERT methods (Sjoberg et al., 2015; You et al., 2017). Rangel et al. (2021) observed taliks below a drained lake basin in the western Arctic Coastal Plain of Alaska using a combination of TEM, NMR, and thermal modeling methods. Yoshikawa et al. (2006) combined six different GPMs (GPR, ERT, TEM, very-low-frequency electromagnetic method, seismic refraction, and helicopter-borne electromagnetic method) to reveal the characteristics of permafrost and taliks in Alaska and the capabilities of these methods were compared. Furthermore, core drilling is often combined with GPMs to help interpret geophysical information (Hilbich et al., 2008; Gao et al., 2019). Despite the contributions made by GPMs toward understanding the permafrost in the QTP, the structure of deep permafrost and sublake taliks remain unclear. Filling this gap can help understand and predict the evolution of the permafrost and the ecological–hydrological environment in the QTP.

This study combined ERT, TEM, and GTM methods to obtain the characteristics of sublake taliks and permafrost structure (spatial distribution and thickness) in the Qinghai–Tibet engineering corridor. ERT and TEM measurements were used to map the permafrost distribution, whereas GTM helped record the thermal state of the sublake taliks and was used to verify the ERT and TEM results. To the best of our knowledge, this is the first study in which GPM and GTM are successfully employed to obtain information on deep permafrost and sublake taliks in the QTP. Overall, the results of this study enrich our understanding of the permafrost and sublake taliks and can provide basic information, such as permafrost thickness and distribution and the morphology of through-taliks, for predicting the development of thermokarst lakes and permafrost and in the maintenance of infrastructure.

## 2 Data and Methods

### 2.1 Study area

The study area is located in the Beiluhe basin in the central QTP, where permafrost and thermokarst lakes are widely distributed (Fig. 1). Important projects, such as the Qinghai–Tibet Railway and Qinghai–Tibet Highway, pass through this basin, making it an important engineering monitoring base (Ke et al., 2022b). The study area has a sub-frigid plateau semi-arid climate, with mean annual precipitation, evaporation, and temperature of 369.8 mm, 1317 mm, and −3.4 °C (Yin et al., 2017), respectively. Rainfall, snow, and hail, accounting for over 80% of the total annual precipitation, mainly occur from June to September. Continuous permafrost with a thickness range of ~50–120 m was formed in cold environments, and the active layer with a thickness range of ~1.6–3.4 m was formed through the periodic freeze–thaw cycle on the ground surface (Ke et al., 2023b). This study was conducted around lakes BLH–A, B, and C (unofficial name) (Fig. 1c). The maximum depth of these lakes is generally 2 m. The areas and maximum lake surface widths of the lakes BLH–A, B, and C are ~1.2, 0.7, and 0.8 ha and ~150, 120, and 135 m, respectively. Lakes BLH–A, B, and C are freshwater lakes with pH and TDS of 8.22–9.71 and 208–952 mg/L,



respectively. Freshwater lakes may have a strong impact on the surrounding permafrost than saline or brackish lakes in the
QTP (Wang et al., 2018; Zhang et al., 2023b), and the saline water may play an important role in holding the bottom of the
lake at a low temperature to preserve the permafrost. $^{210}$Pb and $^{137}$Cs dating showed that the deposition rates of sediment below
lake BLH-A were 0.0056 cm/s and 0.0603 cm/s, suggesting that lake BLH-A may formed 800 years ago (Lin et al., 2010),
respectively. The lakes and the active layer in this region typically start to thaw in mid-May and freeze in mid-October.
Precipitation, supra-permafrost water, and meltwater together help maintain the lake level, which is discharged mainly through
evaporation and surface runoff.

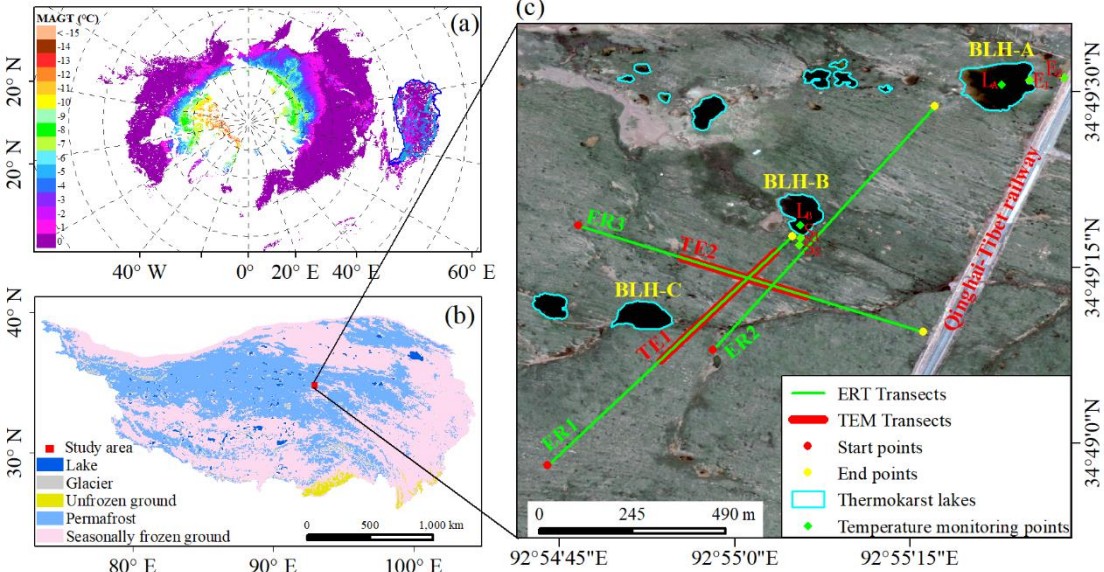

**Figure 1: Map showing the mean annual ground temperature (MAGT) of permafrost regions in the Northern Hemisphere (a) (Ran et al., 2021), permafrost and seasonally frozen ground (permafrost-free below the active layer) distributions of the QTP (b) (Zhao, 2019), and locations of ERT and TEM measurement transects, thermokarst lakes, ground temperature monitoring point (c).**

**2.2 Hydrogeological characteristics**

The study area is located in the topographically flat area of the Beiluhe Basin, which is surrounded by high mountains except
for its eastern part. Surface runoff from precipitation, meltwater, and lake water all flow into the Beiluhe River in the form of
streams. The main landforms are alpine meadows and alpine wetland meadows, and thermokarst lakes are also common natural
landforms (Gao et al., 2021). The stratigraphy of the study area is not very undulating, and the strata are the Quaternary
Holocene alluvial diluvial layer and Neogene lacustrine sedimentary layer from top to bottom (Li et al., 2020). The Quaternary
Holocene alluvial–diluvial layer comprises silt and fine sand layers with a total thickness range of 0.5–3.5 m, and its underlying
brown-red clay layer has a thickness range of 1.0–6.0 m. The Neogene lacustrine sedimentary layer is mainly composed of
strongly weathered mudstone, which is the dominant lithology of the study area (Fig. 2). Based on the borehole information
collected from the Northwest Institute of Eco-Environmental and Resources, CAS, the formation lithology from top to bottom





are sand (~12% water content), silty clay (20%–30% water content), sand (thick-bedded subsurface ice) with a high ice content
(HIC) (more than 50%), and mudstone with a low ice content (LIC) (~17%).

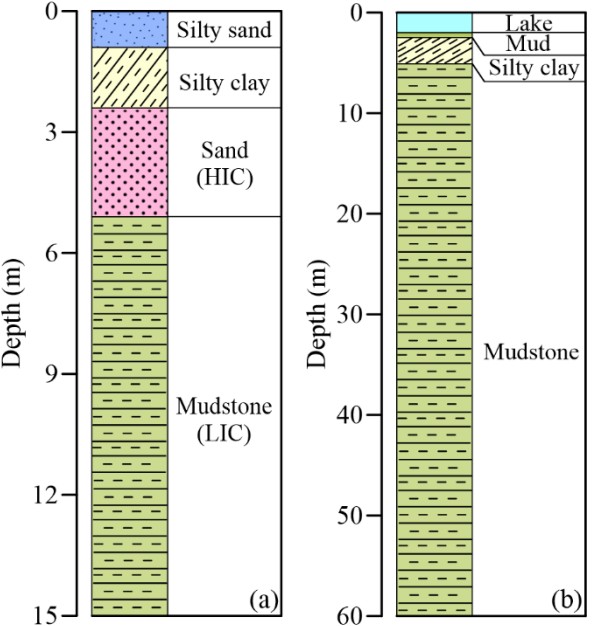

**Figure 2: Stratigraphic information obtained by core drilling in the shore (a) and center (b) of lake BLH–A.**

**2.3 Geophysical surveys**

**2.3.1 ERT surveys and interpretations**

The ERT method, which is based on the electrical differences between geological bodies (rock and soil), can be used to obtain
the distribution of the apparent resistivity ($\rho_s$) by artificially establishing an underground stable current field (Gao et al., 2019).
In this method, current is emitted to the subsurface through electrodes, and $\rho_s$ is calculated from the potential difference
between the electrodes (Zhou and Che, 2021) (Fig. 3a). The distribution of $\rho_s$ is obtained by multiple automatic measurements

between different electrodes; it is then used to infer the distribution of geological elements. For permafrost regions, the
measured $\rho_s$ variations can be attributed to changes in the unfrozen water content, assuming that other conditions (lithology,
pore space, and electrode coupling) are constant (Hilbich et al., 2008). Since $\rho_s$ of unfrozen water is significantly lower than
that of frozen water (Tang et al., 2018), the $\rho_s$ variation in permafrost regions is indicative of the change in the water (ice)
content of the formation. Thus, the permafrost distribution can be inferred from these changes.

Field measurements were conducted in August 2022 in the Beiluhe basin in the central QTP. In this study, the LFGMD-4 ERT
instrument (manufactured by LingFeng Technology Inc., China) (http://www.cqlfgeo.com, last accessed: May 13, 2023) was
used to perform the ER measurements in five transects (numbered ER1 to ER5, referring to Figs. 1 and 4). To analyze the
interaction between the permafrost and thermokarst lake, the ER1 and ER2 transects (Fig. 1) were measured. To reveal the
distribution of the shallow permafrost, the length and electrode spacing of ER1 were adjusted, and ER4 and 5 transects were



obtained (Figs. 4c and 1d). The directions of ER4 and 5 were the same as that of ER1 and close to the end of ER1. To obtain the largely intact permafrost, the ER3 transect was measured far away from the thermokarst lake. The fixed electrode spacings and measuring line lengths for ER1, ER2, ER3, ER4, and ER5 were 10, 10, 10, 5, and 2 m and 790, 790, 790, 195, and 118 m (Table S1), respectively. The elevations of the electrode were also recorded for terrain correction. The industry-standard RES2DINV (Geotomo Software 2011) was employed for the ERT inversion. For the inverse modeling, the smoothness-constrained least-squares method was applied (Loke and Barker, 1996).

### 2.3.2 TEM surveys and interpretations

In the TEM method, electromagnetic fields interacting with the subsurface are generated through a transmitting loop at the ground surface, and a receiving loop is used to measure the decay process of the induced electromagnetic fields over time to estimate the resistivity distribution at various depths (Fig. 3b) (Rangel et al., 2021). The MSD-1 transient electromagnetic instrument (Baiyun Instrument Development Co. LTD., China) was used to collect the ER data from two measuring lines (numbered TE1 and TE2) (Fig. 1). To compare with the ERT results, the directions of the TEM-measured transect were consistent with that of the ERT (ER1 and ER3); however, the lengths were different. The lengths of TE1 and TE2 were 396 and 285 m, respectively. The spacing between the two soundings was 3 m; thus, 132 and 95 soundings were conducted in the TE1 and TE2 transects, respectively. The sounding configuration comprised a square transmitting loop with an area of 40000 $m^2$, a receiving coil with an area of 60 $m^2$ (Table S1), and a main engine (electric current generator). Considering that the lower limit of the permafrost may be deep, the transmitting frequency and stacking fold were set to 25 Hz and 128 (no unit), respectively. The same measurements were performed at 1 m on both sides of TE1 and TE2 to ensure the quality of the collected data. Occam's inversion method was used to interpret the data collected by the TEM method (Constable et al., 1987). A series of 1D interpretation results were interpolated to obtain the 2D results. The workflow for the taliks and permafrost detection can be found in Fig. 3.

### 2.4 Ground temperature monitoring for determining permafrost state

To reveal the thermal state of the thermokarst lake and the surrounding permafrost, the Northwest Institute of Eco-Environment and Resources, CAS, monitored the ground temperature at and around the bottom of the lakes BLH–A and B in 2006 and 2018, respectively. For lake BLH–A, monitoring sites were located at the center of the lake ($L_A$), and 3.9 ($E_1$) and 63.5 m ($E_2$) from the lakeshore (Fig. 1c); the maximum monitoring depths were 58, 15, and 15 m (Table S2) (Lin et al., 2010), respectively. The ground temperatures were automatically monitored using thermistor chains (produced by Northwest Institute of Eco-Environment and Resources, CAS), with errors of less than 0.05 ℃, and two data loggers (DT500, DataTaker., AUS) at 3 h intervals. For lake BLH–B, monitoring sites were set at the center of the lake ($L_B$), and 5 ($S_1$) and 30 m ($S_2$) from the lakeshore (Fig. 1c), with maximum measured depths of 47.5, 5, and 5 m (Gao et al., 2021), respectively. The ground temperatures were automatically monitored using thermistor chains and a data logger (CR3000, Campbell Scientific Inc., USA) at 4 h intervals.



However, only the temperature data of lake BLH–A from 2006/01/09 to 2016/05/05 and of lake BLH–B from 2018/09/02 to 2019/10/26 were collected.

## 2.5 Processes for talik and permafrost detection

Fig. 3c shows the workflow for talik and permafrost detection. A field survey plan was first developed, and the measuring
lines were determined based on the distribution of the thermokarst lake and the basic understanding of permafrost thickness in the study area. Subsequently, borehole data were collected, and the hydrogeological characteristics were analyzed to determine the GPM. Thereafter, the ERT (Fig. 3a) and TEM (Fig. 3b) methods were used to carry out field detection work, and professional software (RES2DINV for ERT) and inversion algorithm (Occam's inversion method for TEM) were used to process and interpret the field data. The distribution characteristics of the permafrost and talik were defined based on the
interpretation results. Finally, the thermal state of the thermokarst lakes and the surrounding permafrost was analyzed using the ground temperature monitoring data, and the geophysical survey results were supplemented.

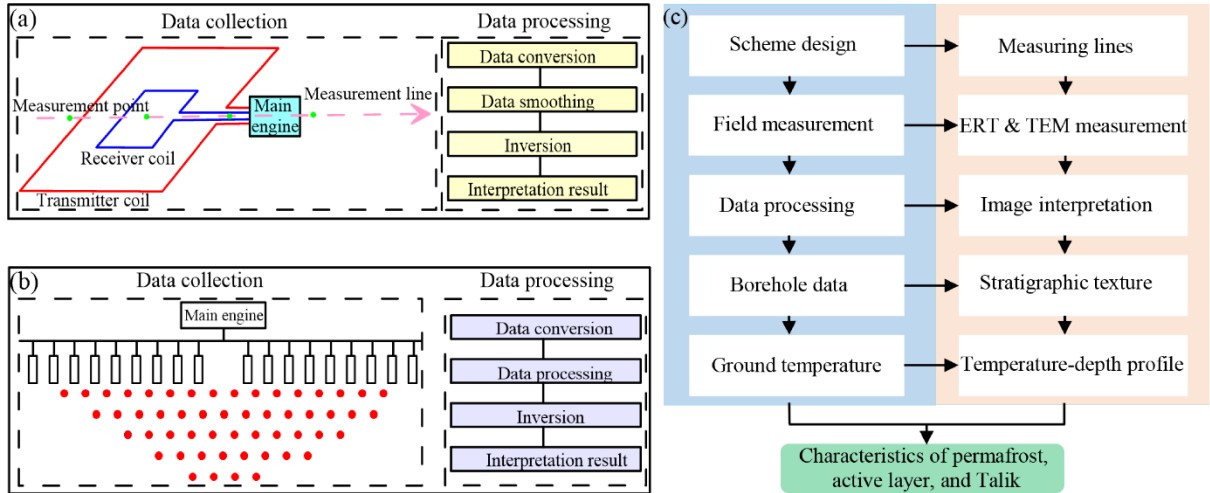

**Figure 3: Schematic and data interpretation process diagram of the TEM (a) and ERT (b) and methods and the framework for permafrost detection (c).**

**3 Results**

## 3.1 ERT geophysical imaging

The ER1 transect commences in the meadows located away from lake BLH–B, traversing a stream in the south of lake BLH–C before passing through a degraded meadow, and finally reaching the shore of lake BLH–B (Fig. 4a). This transect was designed to investigate the impact of lakes, streams, and degraded meadows on the permafrost. All the inversion iterations for
ER1, 4, and 5 were less than 5, and the root-mean-square errors were less than 5.0%, which met the basic inversion requirements. The maximum detection depth for the ER1 transect reached 141 m. The ER of ER1 ranged from 0.7 to 7361




Ω·m with characteristics of high and low values for the shallow and deep layers, respectively (Fig. 4b). Due to the lack of borehole temperature measurements, the variation in ER was used to infer the interface between the permafrost and unfrozen zones. Generally, a rapid decline in ER occurs when transitioning from permafrost to an unfrozen aquifer. The ER (at a distance

of 356.0 m) decreased sharply from 7361 (at a depth of 18.0 m) to 25 Ω·m (at a depth of 51.0 m) and decreased moderately to 10 Ω·m (at a depth of 62.0 m), and then varied from 10.0 to 6.5 Ω·m from a depth of 62.0 m to 141.0 m (Fig. 4b). Similar ER changes can be found at other positions (e.g., at distances of 278, 525, 673 m, etc.); however, the interfaces were at different depths. Therefore, the formation along ER1 could be divided into unfrozen aquifers, an uncertain transitory interface between frozen and unfrozen sediments (Gao et al., 2019), and permafrost with limits of 10 Ω·m and 25 Ω·m based on the ER

distribution. The maximum depth of the permafrost was located under meadows away from the lakes and streams at a depth of 84 m. Seven discontinuous high-ER zones existed within a depth of less than 40 m, which may be a frozen aquifuge with a very low water content. Although lake BLH-C was approximately 45 m from ER1, the lateral and vertical thermal erosion from the lake and stream thawed the deep (similar to that shown in Fig. 4f) and shallow permafrost, respectively. Therefore, it can be inferred that a through-talik had formed below lake BLH-C based on the ERT measurements of ER1 (Fig. 4b) and

borehole temperature measurements in lakes BLH-A and BLH-B (Figs. 9a and b). The sub-permafrost water carrying heat preferentially flowed to zones with higher porosity and permeability, and continuously eroded the permafrost base, causing the permafrost base around the lake to lift significantly. The thermal convection process dominated the permafrost degradation, and its influence on the horizontal was limited; thus, the permafrost-free area increased with increasing depth. The permafrost closer to the lakes BLH–C and BLH–B thawed more sharply, suggesting that these lakes contributed to permafrost degradation.

Under the influence of lake BLH–B, permafrost thickness between lakes BLH–B and BLH–C was approximately 35 m. Notably, the sub-permafrost aquifer was cut off by a band of permafrost with a low ice content or by a formation with a low water content at the right of lake BLH–C. Overall, the through-talik changed the groundwater cycle, which dramatically disturbed the permafrost structure.

ER4 and 5 transects depicted more detailed hydrogeologic information than ER1 within depths of 35 m and 19 m (Figs. 4c

and d), respectively. The ER of ER4 ranged from 9 to 1183 Ω·m with characteristics of high values in the middle layer and low values in the shallow and deep layers (Fig. 4c). The maximum permafrost thickness along ER4 was inferred to be 30 m, which was close to the result along ER1. The thawed active layer increased the water content and decreased the ER of the shallow layer. The maximum upper limit depth of the permafrost was inferred to be 4.0 m. The heat from lake BLH–B and sub-permafrost water thawed the deep permafrost and reduced the ER. ER5 described the ER (ranging from 30 to 1568 Ω·m)

of the active layer (Fig. 4d), and the active layer thickness (ALT) was inferred to be in the range of 0.9–4.0 m. The ALT increased from the starting point to the ending point, which may have been affected by the topography-controlled flow of supra-permafrost water.



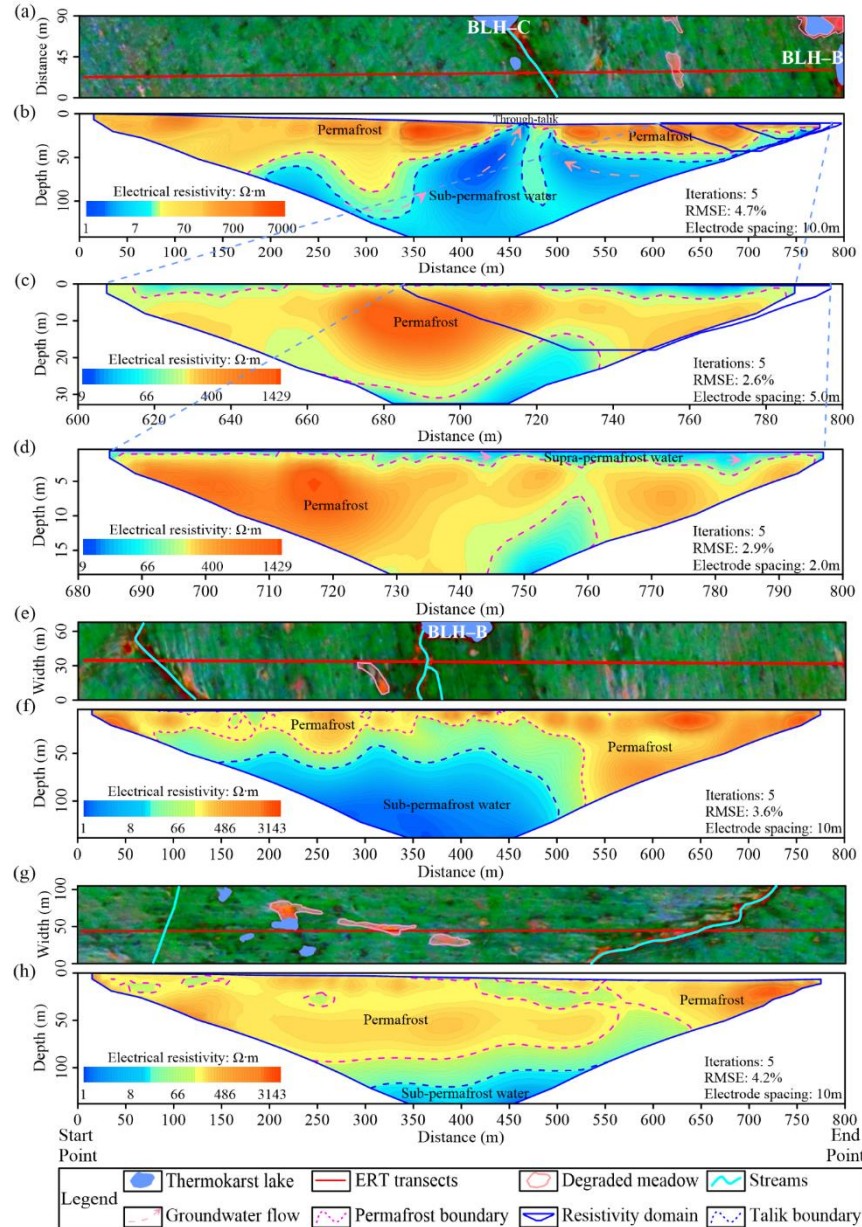

**Figure 4: Land cover along ER1 (a), ER2 (e), and ER3 (g) transects and inversion results of ER1 (b), ER4 (c), ER5 (d), ER2 (f), and**
**ER3 (h). Considering the difference in measurement depths and ER limits, the same color codes are used for ER4 and ER5 and ER2**
**and ER3, respectively; ER1 used a separate color code. The ER domain (continuous blue line) is the boundary of the ER inversion**
**results. The ERs for each profile are relative values rather than true values, and the limits are different; thus, the ER thresholds for**
**the permafrost and talik boundaries for each ERT profile are different.**

ER2 and 3 transects were employed to reveal the effects of lake BLH–B on permafrost (Figs. 4e and f) and the less disturbed

permafrost located far away from the lakes (Figs. 4g and h), respectively. Based on the distribution of the ER, the maximum

permafrost thickness along the ER2 transect was 100 m, located on the right side of lake BLH–B (Fig. 4f). This was due to the



existence of a watershed between lakes BLH–A and B, where groundwater discharged to the two lakes separately, thus having a lesser impact on the permafrost. However, permafrost on the left side of lake BLH–B was less than 50 m thick and incomplete, which was close to the result revealed by ER1 and may have been affected by lakes, degraded meadows, groundwater, and
streams. The ER of the deep layer (below 40 m) in the south of lake BLH–B was very low, suggesting that the deep permafrost was thawed by lake BLH–B. Permafrost in the southern part of lake BLH–B maintained a thickness range of 10–20 m, suggesting the preferential erosion of the permafrost base due to the lateral heat conduction and convection of the lake. For the ER3 transect, the maximum lower limit of the permafrost was inferred to be 93 m (Fig. 4h), which was close to those obtained from ER1 and 2. A sub-permafrost water aquifer was estimated to be present below a depth of 120 m and was relatively evenly
distributed. A relatively low ER zone (in the distance range of 340–550 m) existed in the shallow layer, which may be the thawed permafrost affected by streams and degraded meadows. Notably, some low-ER zones existed in the interior of the permafrost (e.g., in the distance ranges of 34–85, 108–157, and 236–267 m), which may be attributed to lithology or measurement errors. Nevertheless, the ERT method could capture the permafrost and talik structures that were disturbed by the thermokarst lakes, groundwater, and streams.

**3.2 TEM interpretation results**

The electrical resistivity along TE1 and 2 transects were obtained using a 1D inversion calculation based on Occam's inversion theory, which can be used to estimate the electrical resistivity distribution in the subsurface and infer the permafrost and talik structures. To evaluate the accuracy of the inversion results, the induced electromotive forces calculated by Occam's algorithm $((dB_z/dt)_{Cal})$ were compared with the measured data $((dB_z/dt)_{Mea})$ (Figs. 5 and 6). Lower values of the RMSE and higher values
of the correlation coefficient ($R^2$) suggest a better fitting. The $R^2$ and RMSE values for the TE1 transect were 0.978 and $1.78\times10^{-14}$ V·m$^{-2}$, and these values were 0.943 and $3.56\times10^{-14}$ V·m$^{-2}$ for the TE2 transect (Fig. 5), respectively. The minimum slope and maximum intercept of the fitting line were 0.91 and $5.71\times10^{-8}$, which were close to 1 and 0, respectively. The calculated and measured induced electromotive forces exhibited a similar temporal variation, with the lowest $R^2$ and highest RMSE being 0.923 and $6.53\times10^{-14}$ V·m$^{-2}$ (Fig. 6), respectively. Clearly, the calculated values were close to the measured
values, suggesting that the model represents the actual geological environment well, and calculated resistivity can be used to interpret the permafrost structure.





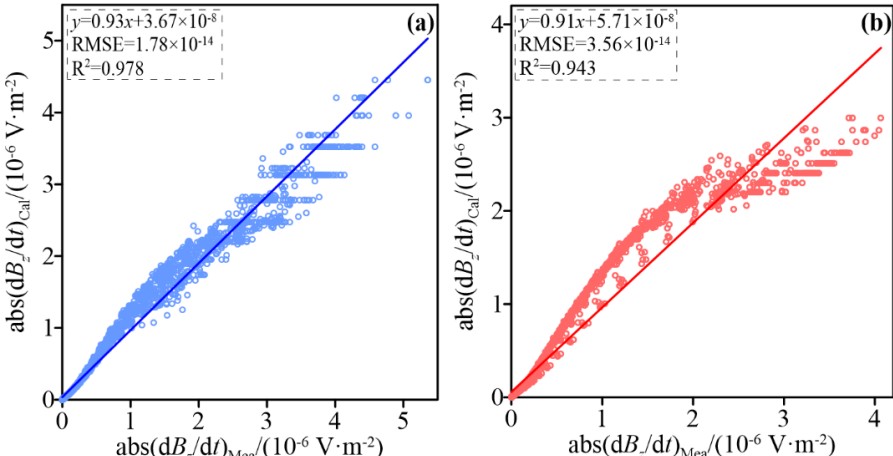

**Figure 5: Comparison between calculated ((dBz/dt)Cal) and measured ((dBz/dt)Mea) induced electromotive forces for TE1 (a) and TE2 (b) transects, where the blue and red lines represent the fitting lines.**

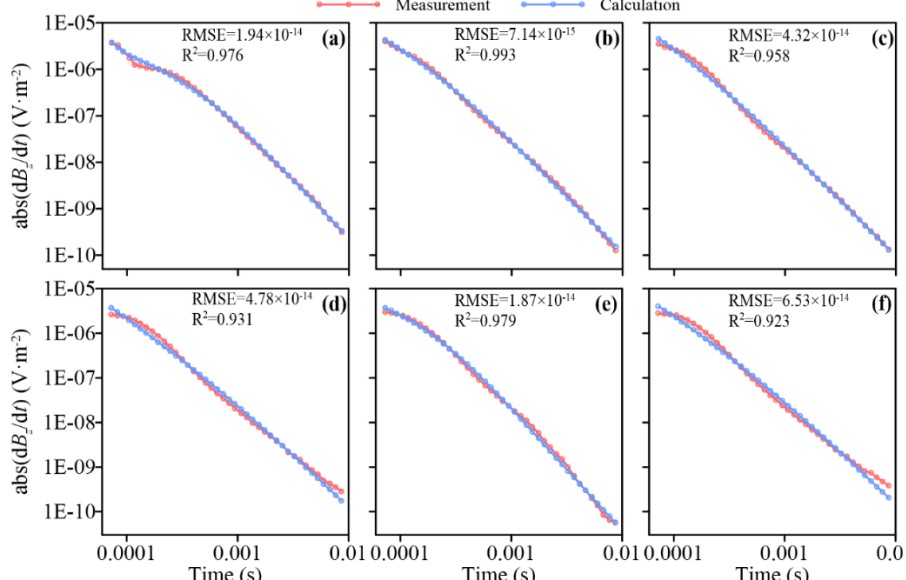

**Figure 6: Variation in measured and calculated induced electromotive forces with time, where a to c and d to f represent the start, middle, and end points for TE1 and TE2, respectively.**

The ER of the TE1 and 2 transects both showed the characteristics of high values in the shallow and deep layers and low values in the middle layer (Fig. 8). The high ER in the shallow and deep layers may represent permafrost and a formation with low water content, respectively, whereas the low ER may indicate the distribution of an unfrozen aquifer (sub-permafrost water). Similar to ERT, changes in ER can be used to determine permafrost and unfrozen aquifer. For the TE1 transect, the rate of ER change (RERC) of the fitting line maintained values less than 0 within a depth of 110 m and greater than 0 at greater depths, indicating that the ER decreased within a depth of 110 m and increased thereafter (Fig. 7). Moreover, the RERC decreased relatively significantly in the depth range of 58–82 m and then began to increase (Fig. 7a), suggesting that the lower limit of



the permafrost may be located within this range. Combined with the ER distribution (Fig. 7a) and the RERC map (Fig. 8a), it can be inferred that the lower limits of the permafrost and aquifer were in the ranges of 40–85 m and 111–150 m, respectively. For the TE2 transect, the RERC was less than 0 in the depth range of 21–112 m, whereas it was greater than 0 in the other detected depth ranges (Fig. 7b), indicating a decrease in the ER in the depth range of 21–112 m and an increase in the other depth ranges. Moreover, the RERC decreased significantly in the depth range of 78–96 m and reached its minimum value, suggesting that the lower limit of the permafrost may be located in the depth range of 78–96 m. Combined with Figs. 7b and 8b, the lower limits of the permafrost and aquifers were estimated to be in the ranges of 83–96 m and 117–142 m (Fig. 8b), respectively.

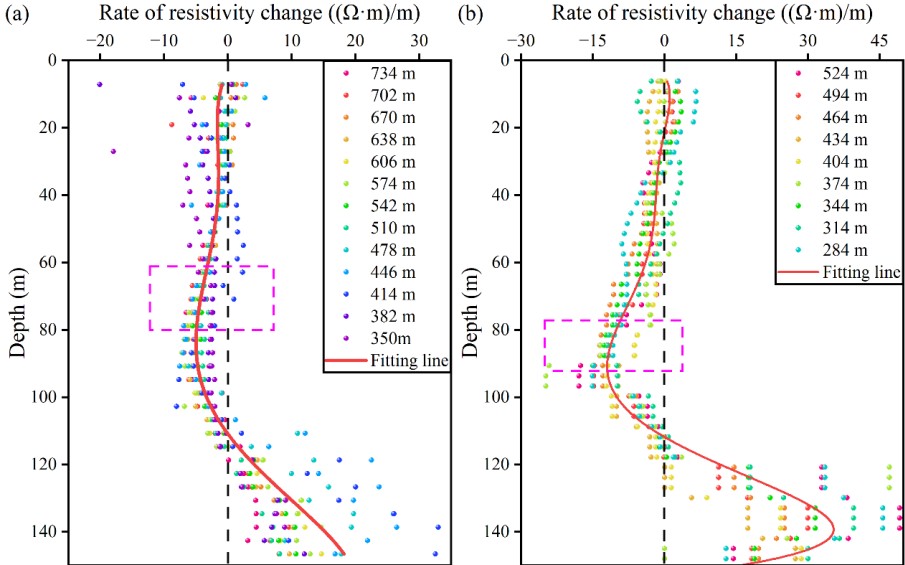

**Figure 7: Rate of resistivity change in TE1 (a) and TE2 (b) transects, where purple dash line boxes mark the possible range of permafrost limits and the black dash lines represent the rate of resistivity change equal to 0.**

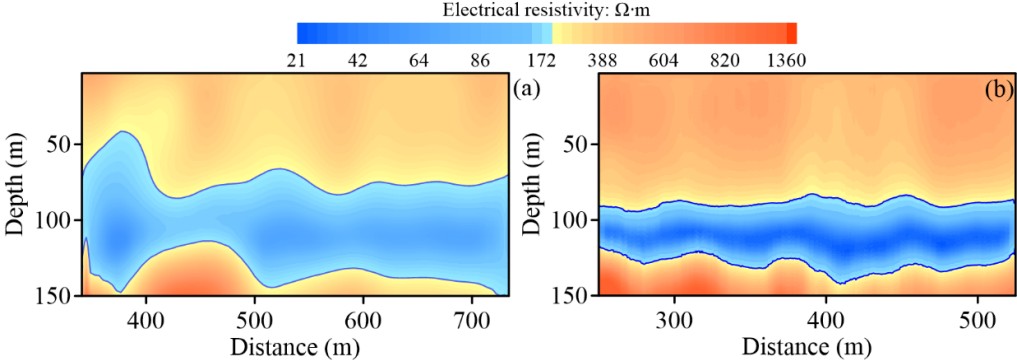

**Figure 8: Contour maps of the ER along the TE1 (a) and TE2 (b) transects, where the blue line indicates the ER limit (200 Ω·m) of the unfrozen aquifers.**




### 3.3 Ground temperature below and around thermokarst lakes

Ground temperature from the thawed active layer to the ice-free zone below the permafrost base first decreased and then increased (Li et al., 2014), because the temperatures of the thawed active layer and ice-free zone (> 0℃) were greater than that of the permafrost (< 0℃). The ground temperatures below the bottom of lakes BLH–A and B were above 0 ℃ (Figs. 9a and b), indicating that the permafrost completely thawed within this depth. Moreover, the temperature below the lake BLH–B decreased first and then increased (Fig. 9b), suggesting the complete disappearance of the permafrost and the formation of a

through-talik. Unlike lake BLH–B, the temperature below lake BLH–A continued to decrease with increasing depth (Fig. 9a), implying the possible existence of permafrost or a permafrost-free zone below the borehole bottom. However, a previous study found that temperature observation data could well record the process of permafrost from its presence to its complete disappearance (Lin et al., 2016), suggesting that a through-talik had also formed below lake BLH–A.

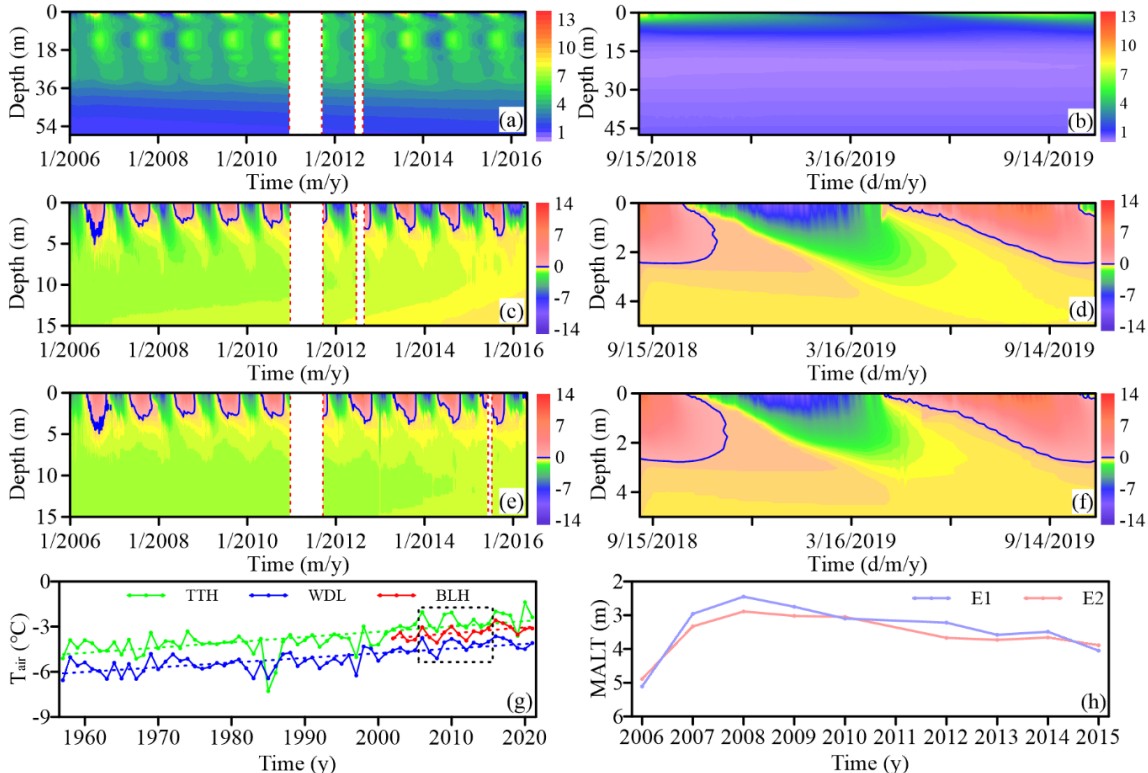

**Figure 9: Contour maps of the ground temperature in borehole $L_A$ (a), $E_1$ (c), $E_2$ (e), $L_B$ (b), $S_1$ (d), and $S_2$ (f), respectively; (g) air temperature in BLH (built in 2002), WDL, and TTH meteorological stations; (h) variation in MALT at $E_1$ and $E_2$, respectively; the areas marked by the red dashed lines (a, c, and e), black dashed line (g) and blue line (c–f) represent missing data, air temperature during 2006–2015, and temperatures equal to 0℃, respectively.**

The mean annual air temperature (MAAT) at Beiluhe (BLH) (2002–2021), Wudaoliang (WDL) (1957–2021) (80 km from

BLH), and Tuotuohe (TTH) (1957–2021) (45 km from BLH) meteorological stations continued to rise at rates of 0.040, 0.031, and 0.035 ℃/a, increasing by 0.70, 2.47, and 2.73℃ (Fig. 9g), respectively, which witnessed climate warming. Under the



influence of climate warming, the temperature of the lake and the ground surface increased, causing the ground temperature to rise gradually (Figs. 9a, c, and e). From 2006 to 2016, the ground temperature contours continuously moved to greater depths (Figs. 9a, c, and e), indicating that permafrost around the lake gradually thawed. A comparison between Figs. 9c and e

shows that the variation in ground temperature was more evident and intense because $E_1$ was closer to lake BLH–A and experienced a stronger thermal erosion. From 2008 to 2016, the upper limit of the permafrost moved to a greater depth (Figs. 9c, e, and h), indicating an increase in the ALT. However, from 2006 to 2008, the MAAT at BLH decreased by 1.02℃ (WDL and TTH decreased by 1.36℃ and 1.27℃, respectively) and the ALT decreased by 2.55 m ($E_1$) and 2.0 m ($E_2$), respectively, indicating that a brief cooling climate decreased the ALT. From 2006 to 2016, the maximum ALT (MALT) (defined by the

0℃ isotherm) values at $E_1$ and $E_2$ were in the ranges of 2.45–5.11 m and 2.89–4.89 m, respectively. The MALT values at $S_1$ and $S_2$ were 2.45 and 2.76 m (Figs. 9d and f), respectively, from September 2018 to September 2019.

**3.4 Comparison and unification of detection results**

The maximum lower limit depths of the permafrost for ER3 and TE2 (in the distance range of 250–525 m) were 93 and 96 m (Figs. 4h and 8b), respectively, suggesting they describe a similar permafrost structure. However, the results of ER1 and TE1

in the distance range of 341–734 m were different, with the maximum lower limit depths of the permafrost for ER1 and TE1 being 34 m and 85 m (Figs. 4b and 8a), respectively. Similar differences were also found for shallow detection along ER3 and TE2. These differences may be attributed to the transmitting frequency (25 Hz) of the TEM. A high transmitting frequency can capture shallow information; however, the lower limit of the permafrost may be difficult to obtain. Therefore, a low transmitting frequency was used, in which case the shallow layer information may be ignored. The ERT method exhibited a

higher resolution and accuracy for shallow layers, and its results were more consistent with the influence of thermokarst lakes, streams, and groundwater on the permafrost. Therefore, the maximum lower limit depths of ER1 or TE1 in the distance range of 341–734 m were determined to be 34 m. Additionally, the maximum lower limit depth of the permafrost for ER2 was 100 m, which was close to those for ER1 (84 m), ER3 (93 m), and TE2 (96 m). As compared above, the maximum lower limit depth of the permafrost (less disturbed) was in the range of 84–100 m. The ALT inferred from the ERT was in the range of

0.9–4.0 m (the average level was 2.45 m), which was close to the result of the borehole temperature ($S_1$ and $S_2$ in September 2019 were 2.45 and 2.76 m, respectively). Based on the ER of ER1 (Fig. 4b) and borehole temperature measurements in lakes BLH-A and BLH-B (Figs. 9a and b), it can be inferred that a through-talik had formed below lake BLH-C. The temperature monitoring results of lake BLH–A indicated the formation of a through-talik below lake BLH–A (Fig. 9a). Similarly, the ground temperatures ($L_B$) and the results of ER2 jointly revealed the complete degradation of the permafrost below lake BLH–

B, forming a through-talik (Figs. 4f and 9b). Overall, the hydrogeophysical investigations clarified the permafrost structure and the effect of thermokarst lakes and groundwater on permafrost.



## 4 Discussion

In this study, GPM and GTM were combined to determine the permafrost and talik structures and to investigate the effect of thermokarst lakes and groundwater on permafrost. Previous hydrogeophysical investigations of the QTP were typically

performed in areas with a small permafrost thickness (e.g., in high plains, basins, and valleys in relatively lower altitude areas) or seasonally frozen soil. Permafrost bases were found at depths of less than 55 m (You et al., 2016; Gao et al., 2019; Yang et al., 2019; Zhou et al., 2022b), whereas in alpine and hilly regions, permafrost bases were typically found at depths greater than 60 m (the maximum value reached 130 m) (Cheng et al., 2019; Gao et al., 2019). Our study was conducted in the Beiluhe basin, which lies in the hinterland of the QTP, and we found that the maximum permafrost depth was in the range of 84–100

m. Furthermore, previous modeling (Li et al., 2021; Ke et al., 2023a) and geothermal gradient analyses (Lin et al., 2010) conducted in our study area showed the existence of permafrost bases at depths of 91 and 85 m, respectively. Gao et al. (2021) found that the unfrozen water content within a depth of 80 m (maximum detection depth) of the lakeshore was low (less than 6%), indicating a wide distribution of thick ice-rich permafrost. As discussed above, the results of the lower limit of the permafrost (permafrost thickness) in this study area are consistent with our findings. Further, previous modeling studies

showed that thermokarst lakes accelerate permafrost degradation, alter groundwater circulation (Wellman et al., 2013; Li et al., 2014; Li et al., 2021), and stimulate the formation of sublake taliks (Creighton et al., 2018; Rangel et al., 2021). Similarly, our study confirmed that the thermokarst lakes continued to transfer heat to the surrounding permafrost, causing the permafrost to thaw and form through-taliks (Li et al., 2021; Ke et al., 2022a). Moreover, the groundwater carrying heat continuously eroded the permafrost base, causing the permafrost base to rise (McKenzie et al., 2007). In particular, the permafrost in the

lakeshore was disturbed by the groundwater and thermokarst lakes, causing the lakeshore to collapse (Niu et al., 2018), thus accelerating lake expansion. These findings highlight the contribution of thermal convection (groundwater flow) toward permafrost degradation (Rowland et al., 2011; Zipper et al., 2018). The formation of through-taliks may cause ecological environmental problems, such as the loss of water resources, water quality deterioration, and vegetation degradation (Niu et al., 2018). Under the backdrop of global warming, the rapid talik expansion and accelerated permafrost degradation exacerbate

these environmental problems (Ding et al., 2019). Our study fills the gap in the investigation of sublake taliks in the QTP and also confirms that thermokarst lakes thaw the surrounding permafrost and form through-taliks (Li et al., 2021; Ke et al., 2022a). The application of GPMs in cryo-hydrogeology can help detect permafrost structures and taliks (Creighton et al., 2018; Rangel et al., 2021). Our study found that ERT can fill the absence of shallow TEM information (depending on the emission frequency: 25 Hz), while the TEM can effectively validate the deep measurement results of the ERT. The synergistic application of the

ERT, TEM, and GTM proved to be an effective and valuable approach for explaining the permafrost and sublake talik structures. The findings of hydrogeophysical investigations on permafrost thickness and the distribution of permafrost and sublake taliks can better guide models in exploring the variation in the symbiotic system of thermokarst lakes and permafrost given their widespread distribution in the Northern Hemisphere.



## 5 Limitations and future work

The findings of this study need to be seen in light of some limitations. First, studies during the cold season and long-term investigations were not performed. Considering that the interaction between thermokarst lake and permafrost is long-term and complex, long-term monitoring will be significant to understanding the process of permafrost degradation and talik development. Second, this study only measured 5 ERT and 2 TEM transects, which may be insufficient for the complex lake–permafrost systems such as the one studied. Third, no new drilling work or temperature measurements were conducted because

of the difficulty and high cost of deep drilling. These limitations may increase the error and uncertainty in the results. Nevertheless, our results revealed the permafrost structure and talik morphologies and the effect of thermokarst lake on permafrost. In the future, more transects and borehole data can be considered for a comprehensive and long-term measurement of the permafrost and taliks. The development of convenient-to-use instruments and measurement methods with high precision, high resolution, and strong applicability can be another research direction.

## 6 Conclusions

It is widely accepted that thermokarst lakes disturb the permafrost. However, in situ detection of the permafrost structure (particularly thick permafrost) and sublake taliks in thermokarst lake regions has been limited. Therefore, in this study, GPMs (electrical resistivity tomography and transient electromagnetic method) and borehole temperature measurements were combined to interpret the permafrost structure and talik morphologies and analyze the effect of thermokarst lakes on permafrost.

The ERT and TEM methods together helped determine the maximum lower limit depth of the permafrost to be in the range of 84–100 m. The ALT was inferred to be in the range of 0.9–4.0 m. The results demonstrated that the permafrost below lakes BLH–A, B, and C had completely thawed and formed through-taliks. Furthermore, the permafrost degradation near the thermokarst lakes was more serious, confirming the contribution of thermokarst lakes to permafrost degradation. Sub- and supra-permafrost water continuously erode the base and top plate of the permafrost, thereby reducing its thickness and

disturbing its structure. The results highlighted the contribution of thermal convection processes to permafrost degradation. The combination of ERT, TEM, and GTM is a useful method for detecting permafrost and sublake talik in cold regions. Our findings can provide necessary constraints (such as permafrost and sublake talik distributions and permafrost thickness) for cryohydrogeologic models of the thermokarst lake–permafrost system to predict permafrost degradation and talik development under the backdrop of future climate warming.

**Declaration of Competing Interest**

The authors declare that they have no known competing financial interests or personal relationships that could have appeared to influence the work reported in this paper.



**CRediT authorship contribution statement**

Xianmin Ke: Conceptualization, Methodology, Software, Investigation, Writing-original draft. Wei Wang: Supervision,
Resources, Methodology, Review & Editing. Fujun Niu: Resources, Methodology. Zeyong Gao: Resources, Methodology.
Wenkang Huang: Data curation & Interpretation. Huake Cao: Data curation & Interpretation.

**Acknowledgments**

This research was funded by the National Natural Science Foundation of China (Grant No. 41730640), the Fundamental
Research Funds for the Central Universities, CHD, Key Laboratory of Eco-hydrology and Water Security in Arid and Semi-
arid Regions of Ministry of Water Resources, Chang'an University Open Fund Funding, and the Talent Support Program of
Hongliu Youqing in Lanzhou University of Technology. We are grateful to Beiluhe Observation and Research Station on
Frozen Soil Engineering and Environment in Qinghai–Tibet Plateau for providing the experimental sites and relevant support.
We thank senior engineer Wentao Liu from the College of Geological Engineering and Geomatics at Chang'an University for
guidance regarding the geophysical prospecting method. Additionally, we are grateful to all the reviewers and editors who
participated in the review as well as to MJEditor (www.mjeditor.com) for its linguistic assistance during the preparation of
this manuscript. All these efforts have helped us improve the quality of the original manuscript.

**Data Availability Statement**

Data used in this study are available from the corresponding author upon reasonable request.

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
