# Peer review of "Thermokarst lakes disturb the permafrost structure and stimulate through-talik formation in the Qinghai-Tibet Plateau, China: A hydrogeophysical investigation"

_EGUsphere, 2025_

## Author Comment (AC1)

**Revisions and responds to reviewer 2's comments**

I appreciate the opportunity to review your manuscript. I think it will make a good contribution to the discipline. Overall, I think the authors present a well-structured study on the impact of thermokarst lakes on permafrost structure in the Qinghai–Tibet Plateau. The combination of geophysical prospecting methods (ERT, TEM) with temperature measurements is a robust approach to investigate sublake talik formation. The manuscript is generally well-written and organized, making it easy to follow the authors' line of reasoning. The topic is relevant, particularly given the context of climate warming and permafrost degradation, and the study appears to be the first of its kind in this specific region. I recommend publication after minor revisions.

*Response: We thank Reviewer 2's positive and encouraging comments which helped us to improve this article considerably. It is our great honor to receive your recommendation. Following are point by point responses to his/her comments. The Reviewer's comments are written in italics and blue and our responses are presented in normal fonts.*

**Specific Comments:**

1. Line 37: The authors state that permafrost degradation is the main driver. However, it should be acknowledged that other factors, such as precipitation and temperature, also play significant roles. Please revise the sentence to reflect this.

*Response: Thanks for your comment. We have revised this sentence to "The surface area and number of thermokarst lakes caused by factors such as permafrost degradation, precipitation, and temperature in the QTP have been increasing (Luo et al., 2022)".*

2. Line 80: The use of "spatial distribution" is too broad. The study focuses on the vertical distribution of permafrost. For accuracy, I recommend changing it to "vertical distribution" or "depth distribution."

*Response: Thanks for your suggestion. We investigated the 2D permafrost structure in 3 transects. Therefore, we used the term "spatial distribution" and declared it.*

3. Line 95: The abbreviation "TDS" is used without prior definition. Please introduce "TDS" (Total Dissolved Solids) at its first occurrence.

*Response: Done and thanks! We have revised this sentence to "Lakes BLH–A, B, and C are freshwater lakes with pH and total dissolved solids of 8.22–9.71 and 208–952 mg/L, respectively".*

4. Line 140: The figure reference should be corrected to read "Figs. 4c and 1d. "

*Response: Thanks for noticing, we have changed it.*

5. Line 189: The streams mentioned to the ERT transect ER1 are not marked in Figure 1. Please add the streams to Figure 1 for clarity. Additionally, consider removing or justifying the inclusion of lakes that are not directly relevant to the study to avoid clutter in Figure 1.

*Response: Thanks for your good suggestion. Following the Reviewer's comments, we removed the irrelevant thermokarst lakes and added streams in Figure 1.*

[Figure]

6. Line 295: The anomaly in ground temperature at a depth of 10-36 m in Figure 9a requires further explanation. Please provide a possible reason for this anomaly. It is critical to address this to give readers confidence in the data's veracity.

*Response: Thanks for your comments. This abnormality may be caused by improper installation of the thermistor or improper handling during drilling and sealing. Although there are local anomalies in the ground temperature data, it can reflect the thermal state below the lake BLH–A and the changes in ground temperature at depth.*

7. Line 297: The text mentions the application value of ground temperature monitoring in cold regions and suggests using the data in Figure 9 to estimate the lower limit of permafrost.

Elaborate on how the temperature gradient can be used for this estimation. This would add value to the discussion.

*Response: Thanks for your comments. Ground temperature monitoring is the most effective method to determine the state of permafrost. In the supplementary information, we have added the calculation of the thawing depth (reach 73.38-87.21 m) below the lake BLH–A using the ground temperature gradient. The corresponding text (To determine whether there was a through-talik below lake BLH–A, the thawing depth (depths of 0°C) below lake BLH–A was estimated by using the geothermal gradient. The results showed the thawing depth was 73.38-87.21 m (Table. S3). With the depth increased, the geothermal gradient decreased. Therefore, the thawing depth may be greater than 87.21 m. The lower limit depth of the permafrost estimated by the borehole temperature was 85 m (Lin et al., 2010). Similarly, a previous study also found that temperature observation data could well record the process of permafrost from its presence to its complete disappearance (Lin et al., 2016), suggesting that a through-talik had also formed below lake BLH–A (formed 800 years ago determined by the $^{210}Pb$ and $^{137}C_s$) have also been added in the manuscript.*

| Selected depth (m) | Temperature at the selected depth (°C) | Ground temperature gradient (°C m$^{-1}$) | Permafrost lower boundary depth (m) |
|---|---|---|---|
| 31.4 | 4.16 | -0.095 | 73.38 |
| 41.4 | 3.05 | -0.084 | 77.67 |
| 51.4 | 2.05 | -0.057 | 87.21 |

8. Figure 9h: The vertical axis of Figure 9h should be reversed to maintain consistency and ease of interpretation.

*Response: Thanks for your comment. We have reversed the vertical axis of Figure 9h.*

[Figure]

9. Abbreviations: A list of acronyms used throughout the manuscript should be included at the end of the paper to improve readability.

*Response: We thank the Reviewer for the good suggestion, which can help improve the readability of the article. We have added an appendix A. Nomenclature list at the end of the paper.*

10. Line 23 in SI: The unit for hydraulic conductivity should be corrected from "(y/m/d)" to "(m/d)".

*Response: Thanks for noticing, we have changed it.*

11. Line 29 in SI: The temporal series data should include the appropriate unit (e.g., temperature in °C).

*Response: Done, thanks for noticing!*

---

## Author Comment (AC2)

**Revisions and responds to reviewer 1's comments**

This is a review of the manuscript titled "Thermokarst lakes disturb the permafrost structure and stimulate through-talik formation in the Qinghai–Tibet Plateau, China: A hydrogeophysical investigation" by X. Ke et al. This manuscript details an investigation primarily focused on geophysical measurements related to permafrost properties around lakes in the Qinghai Tibet Plateau. The main objective is to characterize permafrost structure and the morphology of sublake taliks using direct current electrical measurements and time-domain electromagnetic measurements of permafrost electrical properties. Overall, the text is written clearly and English usage is good. The manuscript lacks a clearly articulated science question, and therefore it is challenging to determine if the main objective of the research was achieved. Additionally, concerns are raised below about geophysical data acquisition, processing, and presentation – while it is not clear that there are detrimental issues with the methods at this point, insufficient information was provided to fully evaluate these issues.

*Response: We sincerely thank Reviewer 1 for taking the time to review our manuscript again and providing valuable suggestions and comments, which helped us to improve this article considerably. We are very sorry that your comments and suggestions were not fully dealt with in the last round of revision. Meanwhile, we also admit that there are some shortcomings in our research and look forward to their resolution in future studies. Here, we have made every effort to respond to your comments and revise our manuscript. Following are point by point responses to his/her comments. Reviewer 1's comments are written in normal fonts and our responses are presented in italics and blue.*

**Specific Comments:**

1. The research question remains unclear. In the manuscript, I was unable to locate the word "question" nor any use of a question mark "?" – a question has not been posed here, and therefore it is difficult to determine if the authors have answered the research question. Furthermore, there is no evidence of a hypothesis that is stated or tested.

*Response: We thank the Reviewer for pointing out the missed research question in the introduction section. Reviewer 3 mentioned this, too. Following the Reviewer's comments, we*

*have added the research questions that need to be solved in this paper at the beginning of the fourth paragraph of the introduction. The corresponding text was rephrased as "Given the widely distributed thermokarst lakes and the paucity of information about permafrost degradation under their influence, we aim to answer the following questions: (1) What is the characteristic of permafrost structure (spatial distribution and thickness)? (2) How do thermokarst lakes affect the permafrost distribution? To answer these questions, we combined ERT, TEM, and GTM methods to obtain the characteristics of sublake taliks and permafrost structures in the Qinghai－Tibet engineering corridor. ERT and TEM measurements were used to map the permafrost distribution, whereas GTM helped record the thermal state of the sublake taliks and was used to verify the ERT and TEM results".*

2. Line 151: The measurement parameters for TEM are confusing. The authors state that a 40,000 m^2 loop was used for transmitting, however, this is an unusually large loop size for such shallow measurements. Typical loop areas for TEM within the top 400 m would be in the range of 1,600 m^2 to 10,000 m^2 (the vast majority being towards the lower end of this range). I was unable to retrieve any information from the manufacturer about this instrument to confirm if 40,000 m^2 is indeed correct, and if so, why such a large loop size would be used in shallow investigations (in the context of TEM, I consider anything <200 m to be a shallow target).

*Response: Thanks for your valuable comments. We understand your concerns regarding the TEM results and acknowledge the limitations of the transmitting loop size used in our field investigation. The 40,000 m² loop was selected to enhance detection capability, as the high resistivity of permafrost was expected to restrict signal penetration. Given the research constraints, we used the MSD-1, an early-generation TEM instrument developed in China, where a larger loop was also necessary to improve signal strength and signal-to-noise ratio. However, data processing revealed a greater investigation depth than initially intended. To obtain reliable shallow subsurface information, we applied a 1D inversion approach and validated the resulting model against the acquired data.*

3. Line 158: The use of the approach in Constable et al 1987 is acceptable, however, the authors do not reference (either in the manuscript or supplement) which codebase or commercial software was used for the inversions. If the authors created their own implementation of the inversion detailed in Constable et al 1987, I would encourage them to share the codebase in

accordance with open data policies and benchmarks should be provided to demonstrate that their code produces consistent results with existing free and paid software that is available. Furthermore, key data pre-processing details are omitted.

*Response: Thanks for your attention to the details of the inversion method and for your valuable comments. We confirm that a 1D inversion based on the method proposed by Constable et al. (1987) was employed in this study. The 1D inversion was carried out in collaboration with Professor Li Xiu's team from Chang'an University, who have developed a dedicated inversion code based on this method. However, following our collaborative agreement, the code is currently not publicly available. We sincerely apologize for this limitation. We fully recognize the importance of code and data transparency in scientific research and appreciate your understanding.*

4. I appreciate that the measured and modeled TEM data are provided in Figure 6 (the same should be done for pseudosections of the ERT data), however these figures seem to reveal that much of the apparent electrical structure has not been fit during the modeling. This is evidenced by the spread of the data and non-linearity of the relationships shown in Figure 5 particularly (but also in Figure 6). In conjunction with the sparse information on TEM pre-processing and inversion make me concerned about the reliability of the TEM results.

*Response: Thanks for your valuable comments. We have used the same processing for pseudosections of the ERT data, and added the relevant figure (provided by RES2DINV software ) and clarifications in the supplementary material and revised manuscript, respectively. Unfortunately, the RES2DINV software cannot export these data for plotting. Therefore, we used and processed pictures printed by RES2DINV, which might result in a relatively low clarity of the pictures.*

*We also sincerely appreciate your continued attention to the reliability of the TEM results. We fully understand your concerns and acknowledge the limitations of our study. In the revision, we have added explanations regarding possible sources of errors and the constraints of our survey strategy. Investigating permafrost structures in the QTP presents significant challenges. To the best of our knowledge, this is the first study in which GPM and GTM are employed to obtain information on deep permafrost and sublake taliks in the QTP. We are grateful that your thoughtful feedback helped reveal these shortcomings, which will be highly valuable for improving future research. Thank you again for your understanding and constructive*

[Figure]

ER1

ER2

[Figure]

ER3

ER4

[Figure]

ER5

*Figure S1: Comparison between calculated and measured apparent resistivities for ER1 to ER5. In the supplementary information, it is presented as two side-by-side images.*

5. Line 190: The method to calculate the maximum detection depth is not stated.

*Response: Thanks for your comment. We acknowledge that the method for estimating the maximum detection depth was not explicitly stated in the previous version. We have clarified this in the revised manuscript. Generally, the maximum detection depth is 1/5 to 1/6 of the transect length (790m), that is, 132 to 158m. The calculated depth of 141 m was determined by RES2DINV based on the electrode configuration and maximum spacing used in the data acquisition. This depth ensures a balance between sensitivity coverage and numerical stability during inversion.*

6. Line 193: "lack of borehole temperature measurements" I don't understand this, ground temperature to >50 m is presented in figure 3, and can easily be modeled to greater depths. Temperature correction should be considered for all geoelectrical images given that large

temperature gradients may be present in the subsurface (e.g., Figure 3), particularly within the top 5-10 meters.

*Response: Thanks for your valuable comment. Temperature gradients can be used to estimate ground temperatures at greater depths. We have deleted this statement. Moreover, we have estimated the thawing depth (depths of 0°C) below lake BLH–A by using ground temperature and temperature gradients, which helped us to judge whether there was a through-talik below lake BLH–A. We fully agree that temperature correction for geoelectrical images is important. Although we wanted to make this beneficial improvement based on your suggestions, we gave up due to the time difference between temperature and resistivity measurements. We will be very glad to carry it out if you have any better suggestions. Nevertheless, we will benefit greatly from these valuable comments in our future work.*

7. Line 324: How was ALT interpreted from this image? It is unwise, if not impossible, to reliably interpret ALT from ERT data because 1) at any reasonable electrode spacing, the ALT will be too close to the surface to image because the layer thickness is ~1x – 4x the electrode spacing and may only occupy 1 or 2 vertical elements in the mesh, and 2) ERT images are inherently smooth and do a poor job of resolving sharp interfaces such as encountered at the ALT.

*Response: We agree with the reviewer that ERT has limitations in resolving shallow features such as the active layer thickness (ALT). To improve near-surface resolution, we used a 2 m electrode spacing. We inferred the ALT using a consistent resistivity threshold that was used in inferring the lower boundary of the permafrost, and the ground temperature information was referred to.*

8. Section 3.3: What is the purpose of this section? It does not appear to play a role in the Discussion section (i.e., Figure 9 is not referenced in the discussion, nor are observations from GTM explicitly discussed in the context of the geophysical measurements and previous research), and if not, why is it included? Presumably the authors would want to consider all of their presented work in the context of other results.

*Response: Thanks for your comments. Ground temperature monitoring is the most effective method to determine the state of permafrost. We hope to determine the permafrost structure and*

*talik by using the thermal state below the thermokarst lake. Moreover, the temperature below the lake BLH–B decreased first and then increased, suggesting the complete disappearance of the permafrost and the formation of a through-talik. Unlike lake BLH–B, the temperature below lake BLH–A continued to decrease with increasing depth, implying the possible existence of permafrost or a permafrost-free zone below the borehole bottom. We have added the calculation of the thawing depth in the supplementary information (reach 73.38-87.21 m) below the lake BLH–A using the ground temperature gradient. As the depth increases, the geothermal gradient decreases. Therefore, the thawing depth may be greater than 87.21 m. The lower limit depth of the permafrost estimated by the borehole temperature (Lin et al., 2010) and by the hydrogeophysical investigation are 85 m and 84-100 m, respectively. Therefore, we infer that a through-talik had also formed below lake BLH–A (formed 800 years ago determined by the $^{210}$Pb and $^{137}$Cs). Additionally, we want to use the temperature data of the active layer to correspond to the thickness of the active layer revealed by the ERT. Therefore, Section 3.3 (Ground temperature below and around thermokarst lakes) existed.*

*Lin, Z., Niu, F., Ge, J., Wang, P., Dong, Y., 2010. Variation characteristics of the thawing lake in permafrost regions of the Tibetan Plateau and their influence on the thermal state of permafrost. Journal of Glaciology and Geocryology 32(2), 341-350. (in Chinese)*

| Selected depth (m) | Temperature at the selected depth (°C) | Ground temperature gradient (°C m$^{-1}$) | Permafrost lower boundary depth (m) |
|---|---|---|---|
| 31.4 | 4.16 | -0.095 | 73.38 |
| 41.4 | 3.05 | -0.084 | 77.67 |
| 51.4 | 2.05 | -0.057 | 87.21 |

9. Section 3.4 may be better suited for the Discussion section, and if moved, should be augmented with appropriate references.

*Response: This is a good suggestion, and we appreciate the reviewer's contributions to enhancing the readability and fluency of the paper. We have moved Section 3.4 to the Discussion section to become the new Section 4.1 and augmented some appropriate references. The revised texts are as follows:*

**4.1 Comparison and unification of detection results**

*The maximum lower limit depths of the permafrost for ER3 and TE2 (in the distance range of*

*250–525 m) were 93 and 96 m (Figs. 4h and 8b), respectively, suggesting they describe a similar permafrost structure. However, the results of ER1 and TE1 in the distance range of 341–734 m were different, with the maximum lower limit depths of the permafrost for ER1 and TE1 being 34 m and 85 m (Figs. 4b and 8a), respectively. Similar differences were also found for shallow detection along ER3 and TE2. These differences may be attributed to the transmitting frequency (25 Hz) used in the TEM survey and simplified inversion model. A high transmitting frequency can capture shallow information; however, the lower limit of the permafrost may be difficult to obtain (Xu, 2014). Therefore, a low transmitting frequency was used, in which case the shallow layer information may be ignored (Zhu et al., 2017). Although inversion can reconstruct geological features to some extent, the simplified model cannot fully capture the complexity of geological structures. Moreover, the TEM inversion models tended to smooth abrupt resistivity changes, leading to smoothed or displaced boundaries. Additionally, the highly heterogeneous geoelectrical structure of permafrost, driven by strong freeze–thaw dynamics, may further amplify discrepancies between inversion results. The ERT method exhibited a higher resolution and accuracy for shallow layers (Li et al., 2021b), and its results were more consistent with the influence of thermokarst lakes, streams, and groundwater on the permafrost. Therefore, the maximum lower limit depths of ER1 or TE1 in the distance range of 341–734 m were determined to be 34 m. Additionally, the maximum lower limit depth of the permafrost for ER2 was 100 m, which was close to those for ER1 (84 m), ER3 (93 m), and TE2 (96 m). As compared above, the maximum lower limit depth of the permafrost (less disturbed) was in the range of 84–100 m. The ALT inferred from the ERT was in the range of 0.9–4.0 m (the average level was 2.45 m), which was close to the result of the borehole temperature ($S_1$ and $S_2$ in September 2019 were 2.45 and 2.76 m, respectively) (Xu et al., 2023). Based on the ER of ER1 (Fig. 4b) and borehole temperature measurements in lakes BLH-A and BLH-B (Figs. 9a and b), it can be inferred that a through-talik had formed below lake BLH-C. The temperature monitoring results of lake BLH–A indicated the formation of a through-talik below lake BLH–A (Fig. 9a) (Lin et al., 2017). Similarly, the ground temperatures ($L_B$) and the results of ER2 jointly revealed the complete degradation of the permafrost below lake BLH–B, forming a through-talik (Figs. 4f and 9b). Overall, the hydrogeophysical investigations clarified the permafrost structure and the effect of thermokarst lakes and groundwater on permafrost.*

*Li, M.N., Zeng, Y.J., Lubczynski, M.W., Roy, J., Yu, L.Y., Qian, H., Li, Z.Y., Chen, J., Han, L., Zheng, H., Veldkamp, T., Schoorl, J.M., Franssen, H.J.H., Hou, K., Zhang, Q.Y., Xu, P.P., Li, F.,*

Lu, K., Li, Y.L., Su, Z.B.: A first investigation of hydrogeology and hydrogeophysics of the Maqu catchment in the Yellow River source region. Earth System Science Data., 13(10), 4727-4757. doi: 10.5194/essd-13-4727-2021.

Lin, Z.J., Niu, F.J., Fang, J.H., Luo, J., Yin, G.A.: Interannual variations in the hydrothermal regime around a thermokarst lake in Beiluhe, Qinghai-Tibet Plateau. Geomorphology., 276, 16-26. doi: 10.1016/j.geomorph.2016.09.035.

Xu, J. The study and application of transient electromagnetic sounding on the theoretical depth of investigation. Master's Thesis, East China University of Technology, Fuzhou, China. (in Chinese)

Xu, Z.D., Jiang, L.M., Guo, R., Huang, R.G., Zhou, Z.W., Niu, F.J., Jiao, Z.P.: Interaction of permafrost degradation and thermokarst lakes in the Qinghai-Tibet Plateau. Geomorphology., 425. doi: 10.1016/j.geomorph.2023.108582.

Zhu, X.G., Fu, Z.H., Su, X.F., Qin, S.Q.: Frequency-domain analysis for pulse current sources in transient electromagnetic method. Near Surface Geophysics., 15(2), 155-162. doi: 10.3997/1873-0604.2016051.

10. Figure 1: It is unacceptable to have the tomograms on different color scales because it makes unbiased interpretation impossible. Each tomogram in this figure must be presented on the same colorscale. Also, I suggest considering the current consensus on colormaps for the presentation of scientific results, and pick one that is more accessible: Crameri, F., Shephard, G. E., & Heron, P. J. (2020). The misuse of color in science communication. Nature communications, 11(1), 5444.

*Response: Yes, the Reviewer is right! We thank the Reviewer for pointing out this error again. We agree with Reviewer 1 that each tomogram in this figure must be presented on the same color scale. Following the Reviewer's comment, we have redrawn the figures and attached them below. First of all, we used the same color scale and the same threshold for all transects. However, for the ER1 transect, the maximum depth of permafrost is less than 40 m (Figure A), which is quite different from the results of borehole temperature, TEM, and other sections. There are significant differences in the range and spatial distribution of their resistivity due to the different environments (water bodies and permafrost distribution) in each transect. We comprehensively considered resistivity and its variations to infer the permafrost boundaries.*

*Therefore, we also consider using the same color scales but with different thresholds for ER1 and other sections (Figure B) to determine the permafrost structure of each transect. We think the second plan is more reasonable.*

*To fully address the Reviewer's concern, we present both options for consideration. While both approaches have merits, we believe that Figure B provides a more accurate representation of the permafrost structure. However, we would greatly appreciate the Reviewer's opinion on this matter and are happy to revise accordingly.*

[Figure]

*Figure A: Inversion results of ER1 (b), ER4 (c), ER5 (d), ER2 (f), and ER3 (h) that used the same threshold.*

[Figure]

*Figure B: Inversion results of ER1 (b), ER4 (c), ER5 (d), ER2 (f), and ER3 (h). The threshold of ER1 is different from that in other sections.*

---

## Author Comment (AC3)

**Revisions and responds to reviewer 3's comments**

The study presents a hydrogeophysical investigation of permafrost and talik distribution around thermokarst lakes in the Qinghai Tibet Plateau. The topic is of interest to me due to the still limited understanding of how permafrost and thermokarst lakes interact, as well as the limited number of studies conducted in the region where this study was performed. The manuscript reads easily and is relatively well organized. I also appreciate seeing the use of TEM for deeper investigation than the ERT, as well as seeing the differences between both methods. However, I have several major concerns, outlined below. I recommend major revisions before publication.

*Response: We thank Reviewer 3 for his/her comments, which helped us to improve this article considerably. Following are point by point responses to his/her comments. Reviewer 3's comments are written in normal fonts and our responses are presented in italics and blue.*

**Major Comments:**

1. The aim of the paper is not clearly stated in the introduction. I suggest that the authors clarify this by introducing a specific research question they are trying to answer or a hypothesis they are aiming to test. Based on the title, I assume the study is focused on the impact of thermokarst lakes on permafrost distribution. Although the discussion returns to this topic, it reads more like a general summary of current literature than a clear comparison of this study's results with previous findings, or a demonstration of how the study confirms or reveals new insights. Additionally, based on the manuscript content, it appears the main focus might be the estimation of permafrost structure around thermokarst lakes, and the comparison between ERT and TEM methods. I suggest the authors refine the key research questions or focus of the study in the introduction, and revise the discussion section accordingly.

*Response: We thank the Reviewer 3 for the excellent suggestion. Reviewer 1 mentioned this, too. As the reviewers mentioned, we focus on the permafrost structure and the impact of thermokarst lakes on it in this article. Following Reviewer 3's suggestion, we have refined the key research questions at the beginning of the fourth paragraph of the introduction. The corresponding text was rephrased as "Given the widely distributed thermokarst lakes and the paucity of information about permafrost degradation under their influence, we aim to answer*

*the following questions: (1) What is the characteristic of permafrost structure (spatial distribution and thickness)? (2) How do thermokarst lakes affect the permafrost distribution? To answer these questions, we combined ERT, TEM, and GTM methods to obtain the characteristics of sublake taliks and permafrost structures in the Qinghai–Tibet engineering corridor. ERT and TEM measurements were used to map the permafrost distribution, whereas GTM helped record the thermal state of the sublake taliks and was used to verify the ERT and TEM results".*

2. It is very difficult to evaluate the ERT and TEM results as presented. The various transects are shown with different resistivity color scales. In Figure 4, ER1 and ER3 intersect, and ER4 and ER5 are subsets of ER1, yet each has a different color/value range. I suggest using a consistent color scale throughout the manuscript, or at least adding all the figure with a consistent color scale in the supplementary material, to allow for meaningful comparison.

*Response: Yes, the Reviewer is right! Reviewer 1 mentioned this, too. Following the Reviewer's comment, we have redrawn the figures and attached them below. First of all, we used the same color scale and the same threshold for all transects. However, for the ER1 transect, the maximum depth of permafrost is less than 40 m (Figure A), which is quite different from the results of borehole temperature, TEM, and other sections. There are significant differences in the range and spatial distribution of their resistivity due to the different environments (water bodies and permafrost distribution) in each transect. We comprehensively considered resistivity and its variations to infer the permafrost boundaries. Therefore, we also consider using same color scales but with different thresholds for ER1 and other sections (Figure B) to determine the permafrost structure of each transect. We think the second plan is more reasonable.*

*To fully address the Reviewer's concern, we present both options for consideration. While both approaches have merits, we believe that Figure B provides a more accurate representation of the permafrost structure. However, we would greatly appreciate the Reviewer's opinion on this matter and are happy to revise accordingly.*

[Figure]

*Figure A: Inversion results of ER1 (b), ER4 (c), ER5 (d), ER2 (f), and ER3 (h) that used the same threshold.*

[Figure]

*Figure B: Inversion results of ER1 (b), ER4 (c), ER5 (d), ER2 (f), and ER3 (h). The threshold of ER1 is different from that in other sections.*

3. The use of color scale is also misleading in Figure 8, I think. It gives the impression of a sharp subsurface boundary without showing the actual "smoothed" inversion result. If the authors want to show results in this manner, I believe the inversion should first be presented with a more typical color scale at least in the supplementary material for transparency. In addition, it is very likely that there are discrepancies btw the resistivities values and boundaries identified with the TEM and ERT. I do believe that improving the discussion of differences and reasons (in section 3.4 or section 5) would benefit the paper.

*Response: Yes, the Reviewer is right! Subsurface boundary should be smoothed, like the profile (AA' and BB') captured in the contour map. To show the differences between frozen and unfrozen strata more clearly, we changed the color scale and transparency and added some contour lines in Figure 8.*

[Figure]

*Following the Reviewer's comment, we have added the clarification for the discrepancies between the resistivity values and boundaries identified with the TEM and ERT. The revised text is as follows:*

*These differences may be attributed to the transmitting frequency (25 Hz) used in the TEM survey and simplified inversion model. A high transmitting frequency can capture shallow information; however, the lower limit of the permafrost may be difficult to obtain. Therefore, a low transmitting frequency was used, in which case the shallow layer information may be ignored. Although inversion can reconstruct geological features to some extent, the simplified model cannot fully capture the complexity of geological structures. Moreover, the TEM inversion models tended to smooth abrupt resistivity changes, leading to smoothed or displaced boundaries. Additionally, the highly heterogeneous geoelectrical structure of permafrost, driven by strong freeze–thaw dynamics, may further amplify discrepancies between inversion results.*

**Specific Comments:**

1. Line 127: Apparent resistivity ($\rho_s$) is not the same as resistivity. In this paragraph, the term "apparent resistivity" seems to be used incorrectly or inconsistently. Consider clarifying by discussing the measured resistance and the inferred electrical resistivity separately.

*Response: Thanks to the Reviewer for indicating this incorrect and inconsistent term. We originally regarded the resistivity measured by ERT and calculated by inversion as apparent resistivity and resistivity, respectively. Following the Reviewer's comment, we have changed apparent resistivity ($\rho_s$) to measured resistance and unified the inversion resistivity as electrical resistivity (ER). The revised text is as follows:*

*The ERT method, which is based on the electrical differences between geological bodies (rock and soil), can be used to obtain the distribution of the measured resistance by artificially establishing an underground stable current field (Gao et al., 2019). In this method, current is emitted to the subsurface through electrodes, and measured resistance is calculated from the potential difference between the electrodes (Zhou and Che, 2021) (Fig. 3a). The distribution of measured resistance is obtained by multiple automatic measurements between different electrodes; it is then used to invert the distribution of the ER and infer the geological elements. For permafrost regions, the ER variations can be attributed to changes in the unfrozen water content, assuming that other conditions (lithology, pore space, and electrode coupling) are constant (Hilbich et al., 2008). Since ER of unfrozen water is significantly lower than that of frozen water (Tang et al., 2018), the ER variation in permafrost regions is indicative of the change in the water (ice) content of the formation. Thus, the permafrost distribution can be inferred from these changes.*

2. Line 173: Consider moving Section 2.5 earlier in the manuscript to explain the survey strategy before providing details about each method. This will help the reader understand the overall approach more clearly.

*Response: This is an excellent suggestion, and we appreciate the reviewer for making our manuscript more fluent and readable. Following the Reviewer's comment, we have moved Section 2.5 (Processes for talik and permafrost detection) after Section 2.2 (Hydrogeological characteristics) to make it the new Section 2.3.*

3. Line 227: The statement "The ERs for each profile are relative values rather than true values..." is confusing. Electrical resistivity is not a relative value. Please clarify what is meant here.

*Response: Yes, the Reviewer is right! We thank the Reviewer to point out this confusing statement. We have deleted this sentence.*

4. Figure 5: Consider explaining why the fit appears low. If the misfit were expressed as a percentage, I believe it would appear significant. Please consider providing the misfit in % and discussing the possible reasons more clearly.

*Response: Yes, the Reviewer is right! The misfit in % can show the difference between the measured and simulated values. The mean relative error (MRE) of the TE1 and TE2 transects have been added in Figure 5, which were 9.35% and 17.64%, respectively. Furthermore, we also calculated the REs of the average values of the TE1 and TE2 transects, which were 0.07% and 2.63%, respectively. We have explained the possible reasons for the errors in the manuscript and the corresponding text is as follows:*

*The calculated and measured induced electromotive forces exhibited a similar temporal variation, with the lowest $R^2$ and highest RMSE being 0.923 and $6.53 \times 10-14$ V·m$^{-2}$ (Fig. 6), respectively, suggesting that calculated values were close to the measured values. Nevertheless, there were deviations between the measured and simulated values, with the mean relative errors (MREs) of TE1 and TE2 being 9.35% and 17.64%, respectively. There may be two main reasons for these errors. First, the simplified model cannot fully reflect the complex geological conditions. Permafrost has a strong spatial heterogeneity in electrical conductivity due to the uneven mixture of ice, water, and minerals. This causes local enhancement or attenuation of electromagnetic fields, which simplified models difficult to capture–particularly near electrical conductivity anomalies like permafrost-aquifer boundaries, where errors were more significant. Second, electromagnetic responses in high-resistivity permafrost were weak, further contributing to discrepancies between the model and actual observations.*

[Figure]

5. Line 364: It is great to see a section dedicated to "Limitations." However, I am surprised there is no discussion of challenges related to data inversion in TEM, the comparison of ER values between ERT and TEM, or the difficulty in defining a boundary value to delineate permafrost.

*Response: Yes, the Reviewer is right! Thanks for noticing, we only focused on the limitations of the field investigation. This is the first attempt to obtain information on deep permafrost and sublake taliks in the QTP using GPM and GTM. Therefore, there are inevitably some limitations and challenges regarding field investigation strategies, geophysical exploration applications, and methodology, which are hoped to be further overcome in future studies. Following the Reviewer's comment, we have added the shortcomings related to data inversion in TEM, the comparison of ER values between ERT and TEM, and the determination of permafrost boundaries. The revised text is as follows:*

*The findings of this study need to be seen in light of two major limitations. One is regarding the field investigation. Studies during the cold season and long-term investigations were not performed. Considering that the interaction between thermokarst lake and permafrost is long-term and complex, long-term monitoring will be significant to understanding the process of permafrost degradation and talik development. Moreover, only measured 5 ERT and 2 TEM transects, which may be insufficient for the complex lake–permafrost systems such as the one studied. Additionally, no new drilling work or temperature measurements were conducted because of the difficulty and high cost of deep drilling. Another limitation concerns the GPMs themselves. TEM inversion is based on a simplified model that may not adequately reflect the*

*complex subsurface environment in permafrost regions. Discrepancies in ER between ERT and TEM make it difficult to direct comparison. Moreover, defining a consistent ER threshold to delineate permafrost boundaries remains challenging, as ER is controlled by factors such as ice or water content and rock type. These limitations that need to be overcome increase the error and uncertainty in the results. Nevertheless, our results revealed the permafrost structure and talik morphologies and the effect of thermokarst lake on permafrost. In the future, more transects and borehole data can be considered for a comprehensive and long-term measurement of the permafrost and taliks. The development of more applicable calculated methods and convenient-to-use instruments with high precision, high resolution, and strong applicability can be another research direction.*

---

## Author Response (AR2)

**Revisions and responds to editor and reviewers' comments**

First of all, we would like to thank the Editor and two reviewers for their comments and suggestions, which greatly improved the presentations and interpretations in our revised manuscript. In the revised article, we have addressed all comments and suggestions from the Editor and two reviewers. All changes made in the manuscript are in red. Our point by point response to the Editor and two reviewers' comments is outlined below. The original comments are shown in normal fonts and responses are given in italics and blue. We hope the revised manuscript will meet the journal's standards.

**Editor:**

Thank you for submitting your revised manuscript. As stated by the two reviewers, the paper has considerably improved. Nevertheless, I share the concerns of one of the reviewers concerning the TEM data. I also agree that the signals are actually not that weak, and you actually show a good agreement between ERT and TEM data. On the other hand, I agree that permafrost systems are very complex, and that a simple 1D model may not explain the variability you have in the subsurface. However, it would be good to either extend the discussion on that, or as the reviewer suggests that if the added value of the TEM measurements is low, to just remove them and focus on the ERT data.

Response: We are grateful to the Editor for taking the time to handle our manuscript and giving us valuable comments. As you and the reviewer were concerned, the TEM has some limitations in equipment, signal acquisition, and data interpretation. However, at present, we have no better solution to overcome these problems, either in data processing or in modeling. Following your and the reviewers' suggestions, we have removed all the information related to the TEM and focused instead on the results of the ERT. We have made every effort to address all comments and suggestions from the Editor and two reviewers, and sincerely hope that the revised article can meet the journal's standards.

**Reviewer #1**

The substantial revisions completed by the manuscript authors are recognized. Many of the revisions have provided meaningful improvements (e.g., addition of a research question) while others fell short of expectations (e.g., declined to apply temperature correction to ERT images).

Response: We thank the Reviewer's positive comments and encouragement which help us to improve this article considerably. The comments regarding the application of temperature to correct ERT images are highly valuable and insightful. However, inconsistencies between the temperature monitoring locations and periods and the ERT measurements present challenges for implementing such corrections. We greatly value this constructive suggestion. While we have not yet found a suitable solution to overcome this limitation in the current work, the comment provides valuable guidance for future research.

Line 31: "Permafrost is a special type of sediment..." permafrost may be any material – sediment or otherwise- that remains below 0C for 2 or more consecutive years.

Shur, Y., Jorgenson, M. T., & Kanevskiy, M. Z. (2011). Permafrost. In Encyclopedia of snow, ice and glaciers (pp. 841-848). Springer, Dordrecht.

Response: We thank the Reviewer to point out this imperfect statement. Following the Reviewer's comment, we have rephrased text as "Permafrost is ground (soil or rock and included ice and organic material) that remains at or below 0°C for at least 2 consecutive years (Gao and Coon, 2022; Shur et al., 2011)..." (line 31-32)

Line 255-285: The use of the TEM data remains concerning – after the detailed comments related to the TEM data by multiple reviewers/comments, the authors did not attempt to improve the fit or otherwise explore the inversion of the dataset, and rather provided only two possible explanations: 1) the permafrost is heterogeneous, and 2) the EM responses of resistive permafrost are weak. For 1, this may be the case (as evidenced in part by the ERT data), but if it is in fact not possible to resolve the complex structure with the TEM data, why use this dataset at all, knowing (based on ERT) that the image is fundamentally incorrect? For 2, this is physically true, however the inverted results of both EM and ERT indicate a strong conductor at 80-120m that would be expected to produce strong signals. Therefore, such an argument about low signal strength does not seem to be valid. Furthermore, the comparison between the

TEM and ERT, as described in the first paragraph of the Discussion section, is inconsistent. I suggest the TEM dataset be removed from the manuscript and focus instead on the ERT data. Response: Thanks for your comments. We understand the Reviewer's concerns. The TEM has some limitations in equipment, signal acquisition, and data interpretation that we have no better solution to overcome presently. Following the reviewers' suggestions, we have removed all the information related to the TEM and focused instead on the results of the ERT.

Line 334: It would be helpful to have a Discussion sub-section dedicated to answering the newly posed research questions, in particular question #2 related to how permafrost structure determines distributions of thermokarst lakes, which has not been answered in the current version of the discussion.

Response: Yes, the Reviewer is right! Following the Reviewer's comment, we have added a Discussion sub-section to answer question #2 related to how thermokarst lakes affect the permafrost distribution. The added texts are as follows:

**4.3 The effect of the thermokarst lake on permafrost structure**

In regions without thermokarst lakes, permafrost dynamics are mainly controlled by air temperature, precipitation, and groundwater flow, which primarily influence the active layer; thus, the permafrost structure remains relatively stable (Fig. 4h) (Wu et al., 2022). However, the temperatures of thermokarst lakes are generally higher than those of the surrounding permafrost, thereby accelerating permafrost degradation (in 't Zandt et al., 2020). In the initial stage of thermokarst lake development, permafrost impedes groundwater flow, and heat transfer is dominated by thermal conduction (McKenzie et al., 2007). Under these conditions, thawing mainly occurs vertically, concentrated beneath the lake bottom and within limited areas around the lake (Fig. 4b) (Niu et al., 2018). As thawing progresses and permafrost below the lake bottom disappears, a through-talik forms, which restores groundwater flow pathways. Groundwater carrying heat preferentially migrates through highly permeable zones, resulting in progressive thawing at the base of the permafrost (Figs. 4b and f) (Li et al., 2021a). Our investigation revealed substantial thawing at the permafrost base near the lake, indicating that groundwater plays a critical role in the thawing process, which highlights the significance of

thermal convection in permafrost degradation and confirms previous numerical simulation results (Rowland et al., 2011; Zipper et al., 2018). In particular, the permafrost in the lakeshore was disturbed by the groundwater and thermokarst lakes, causing the lakeshore to collapse (Niu et al., 2018), thus accelerating lake expansion. Furthermore, in regions with a thermokarst lake group, permafrost between adjacent lakes exhibited pronounced degradation, with a remarkable thinning of its thickness after through-talik formation (Figs. 4b, c, and d) (Ke et al., 2023a). In summary, thermokarst lakes disturb permafrost structure through two primary mechanisms: (1) direct thermal erosion caused by relatively warm lake water, and (2) alterations in groundwater circulation that enhance subsurface thaw (the main mechanism). These processes emphasize the combined influence of thermokarst lakes and groundwater on permafrost dynamics in permafrost regions. These findings enhance our understanding of the interaction mechanisms between thermokarst lakes and permafrost, and contribute to research on the evolution of thermokarst lakes and their associated ecological and environmental impacts in cold regions. (lines 309-330)

**Reviewer #2**

Response: We sincerely thank Reviewer #2 for taking the time and effort to review our manuscript. It is our great honor to receive your recommendation.